# Intragranular cracking as a critical barrier for high-voltage usage of layer-structured cathode for lithium-ion batteries

Pengfei Yan[1,*], Jianming Zheng[2,*], Meng Gu[1], Jie Xiao[2], Ji-Guang Zhang[2] & Chong-Min Wang[1]

$LiNi_{1/3}Mn_{1/3}Co_{1/3}O_2$-layered cathode is often fabricated in the form of secondary particles, consisting of densely packed primary particles. This offers advantages for high energy density and alleviation of cathode side reactions/corrosions, but introduces drawbacks such as intergranular cracking. Here, we report unexpected observations on the nucleation and growth of intragranular cracks in a commercial $LiNi_{1/3}Mn_{1/3}Co_{1/3}O_2$ cathode by using advanced scanning transmission electron microscopy. We find the formation of the intragranular cracks is directly associated with high-voltage cycling, an electrochemically driven and diffusion-controlled process. The intragranular cracks are noticed to be characteristically initiated from the grain interior, a consequence of a dislocation-based crack incubation mechanism. This observation is in sharp contrast with general theoretical models, predicting the initiation of intragranular cracks from grain boundaries or particle surfaces. Our study emphasizes that maintaining structural stability is the key step towards high-voltage operation of layered-cathode materials.

[1] Environmental Molecular Sciences Laboratory, Pacific Northwest National Laboratory, 902 Battelle Boulevard, Richland, Washington 99352, USA. [2] Energy and Environment Directorate, Pacific Northwest National Laboratory, 902 Battelle Boulevard, Richland, Washington 99352, USA. * These authors contributed equally to this work. Correspondence and requests for materials should be addressed to J.-G.Z. (email: Jiguang.zhang@pnnl.gov) or to C.-M.W. (email: Chongmin.wang@pnnl.gov).

Exploring lithium-ion battery (LIB) electrode degradation mechanisms has long been an active research topic for the battery community[1–16]. Understanding the origin of degradation allows us to design better electrode materials. In the case of layered transition metal (TM) oxide cathode degradation, three mechanisms have been identified[2–5,12,17,18]: (1) Layer to spinel/rock salt phase transformation, which is characteristically initiated from the individual particle surface and gradually propagated inwards with battery cycling. (2) Side reactions between the cathode and electrolyte, leading to electrolyte decomposition and passivation of the solid electrode. (3) Corrosion and dissolution of the cathode materials in the electrolyte. These findings lead to the application of coating techniques and other surface treatments to stabilize vulnerable surfaces on the cathode materials. Such coating and surface treatments have been frequently verified as effective methods for improving cathode cycling stability[19–28]. Besides chemical

instability, another degradation mechanism is associated with the volume change of the material upon lithium (Li) ion extraction and reinsertion. Non-uniform accommodation of such a volume change will generate stress, which can lead to mechanical failure[29]. In fact, intergranular crack formation is one of the most well-known material degradation mechanisms[29–35].

During the charge process of layered TM oxides, Li ions are extracted from the lattice, which usually causes lattice expansion along the $c$ direction and shrinkage along the $a$ and $b$ directions[29,36–38], which is reversed upon discharging. This type of lattice expansion and shrinkage is generally termed as lattice breathing, which has been theoretically and experimentally verified. For example, when $LiNi_{1/3}Mn_{1/3}Co_{1/3}O_2$ (NMC333) is delithiated to $Li_{0.5}Ni_{1/3}Mn_{1/3}Co_{1/3}O_2$, Yoon et al.[37] found that the lattice could expand 2.0% along the $c$ direction and shrink 1.4% along the $a$ direction, inducing significant strain for oxides with ionic bonds.

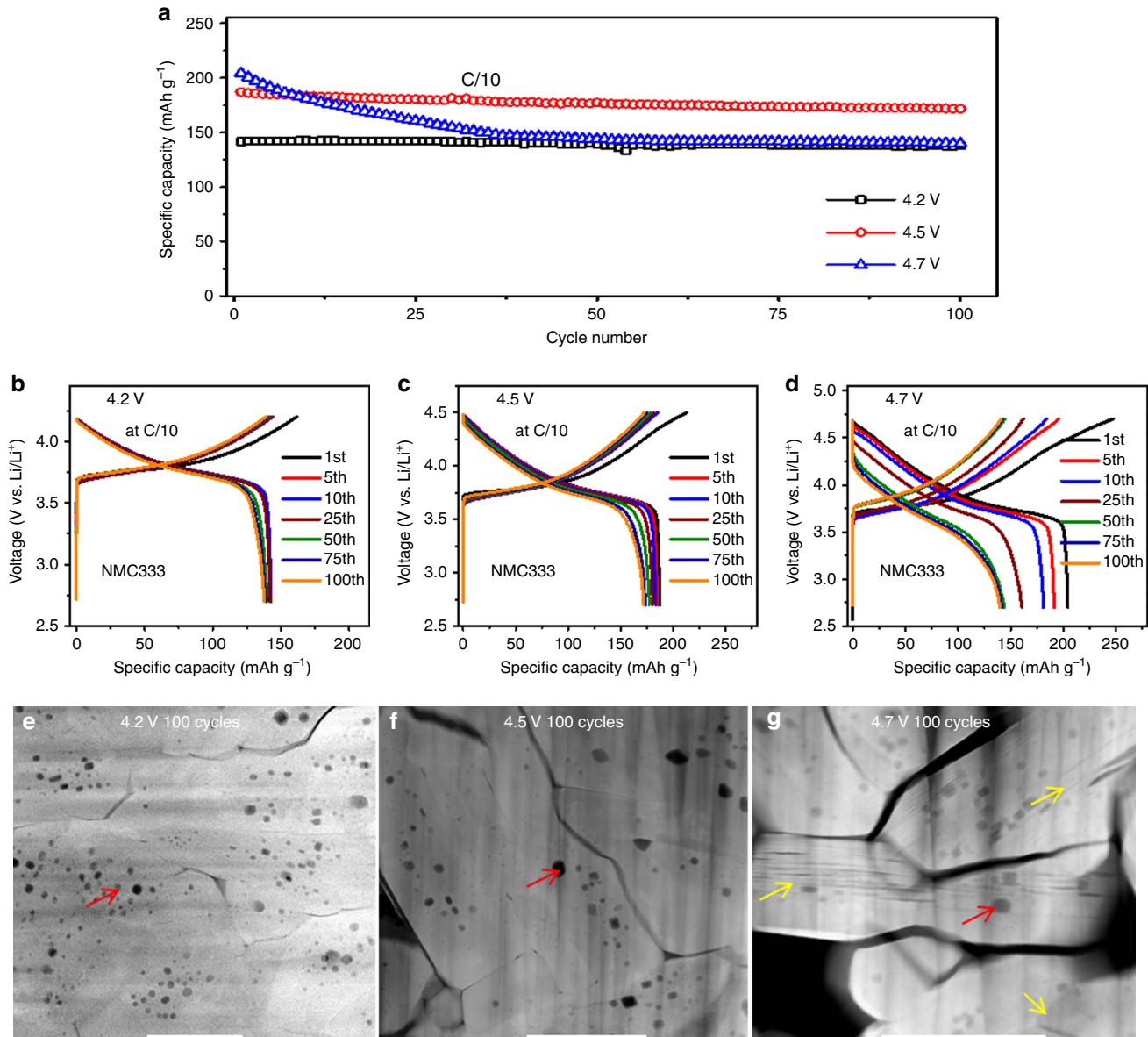

Figure 1 | Electrochemical performance and observations of fracture. (a) Specific capacity as a function of cycle number, revealing the $Li/LiNi_{1/3}Mn_{1/3}Co_{1/3}O_2$ half cell's capacity fading has strong dependence on the high cutoff voltages, (b–d) charge/discharge profiles of $Li/LiNi_{1/3}Mn_{1/3}Co_{1/3}O_2$ half cells at different high cutoff voltages, and (e–g) low magnification HAADF images of $LiNi_{1/3}Mn_{1/3}Co_{1/3}O_2$ after 100 cycles at different high cutoff voltages. The red arrows indicate voids and the yellow arrows in g indicate intragranular cracks. Scale bars, 500 nm (e–g).

To increase the volumetric energy density of the electrode, the packing density of the active electrode component should be increased. One way to accomplish this in commercial LIB cathodes is to use primary particles to form densely packed secondary particles. However, such secondary particles always generate intergranular cracks during battery charge/discharge cycling, due to the anisotropic expansion and shrinkage of each primary particle[31,32,39]. Such strain-induced cracking has been considered to be one of the major degradation mechanisms for the cathode for the following reasons: (1) Cracks can result in poor grain-to-grain connections, leading to poor electrical conductivity and even loss of active materials due to fragmentation; (2) Cracks create fresh surfaces that will be exposed to electrolytes and generate new sites for surface phase transformation, corrosion and side reactions, consequently accelerating cell degradation.

Besides intergranular cracks, intragranular cracks were also observed in several cathode materials after prolonged cycling[39–41]. For example, Chen et al.[40] found cracks in the bc planes of $LiFePO_4$, Wang et al.[41] noted cracks in $LiCoO_2$ particles, and Kim et al.[39] observed cracks in $LiNi_{0.6}Mn_{0.2}Co_{0.2}O_2$ after 150 cycles at $60\,^\circ C$. Compared with intergranular cracks, intragranular cracks are smaller in size but much higher in density, and thus, they can generate many more fresh surfaces that will be exposed to electrolytes. Moreover, intragranular cracking is not only a mechanical failure but also more likely to be a structural degradation under severe electrochemical conditions. Therefore, the previously proposed effective design concepts (such as surface coating) for preventing intergranular cracking[31,39,42,43] may not solve the intragranular cracking problem. Mitigation of intragranular cracking requires a stable structural framework of the cathode material and careful controls of cycle conditions. A systematic investigation on intragranular cracking in cathode materials is still lacking.

In this work, we report detailed observations on the cracking phenomenon in NMC333 layered-cathode materials by using advanced scanning transmission electron microscopy (STEM). In particular, the intragranular cracking process is comprehensively investigated. We find that the density of intragranular cracks in NMC333 cathodes abruptly increases when cycled at a high cutoff voltage of 4.7 V. In contrast expectations, we also observe the intragranular cracks to actually initiate from the grain interior, which is in sharp contrast with general theoretical models predicting the surface or grain boundary to be the preferred sites for intragranular crack initiation[42,44–46]. We also verify that the edge dislocation core can assist the incubation of intragranular cracks, and that intragranular cracking is an electrochemically driven and diffusion-controlled process, mimicking the classic model of slow crack growth during fatigue process of materials.

## Results

**High cutoff voltage cycling induced intragranular cracking.** The performance of NMC333 cathode electrodes in Li/NMC333 half-cells cycled at different voltage ranges, that is, $2.7 \sim 4.2$ V, $2.7 \sim 4.5$ V, and $2.7 \sim 4.7$ V, are shown in Fig. 1a, and the corresponding charge/discharge voltage profile evolutions are shown in Fig. 1b–d, respectively. Electrochemical data indicate that the cycling stability of the NMC333 cathode shows strong dependence on the charge cutoff voltages that are applied for battery cycling. Generally, the higher the charge cutoff voltage, the faster degradation of the battery performance. When the battery was cycled at a low charge cutoff voltage of 4.2 V, the NMC333 shows excellent cycling stability along with very limited voltage decay. However, with the increase of charge cutoff voltage, the NMC333 shows obvious voltage fading and capacity decay. Particularly,

serious voltage decay and capacity fading occur when cycling at 4.7 V.

Because excessive Li metal was used as anode for these three cells, it is believed that their performance difference should be mainly associated with instability of the cathode and electrolyte. It is known that a higher charge voltage can result in aggravated degradation of the cathode and electrolyte due to the side reactions between the cathode and electrolyte, formation of a thicker phase transformation layer on the surface of the cathode, and severe surface corrosion of the cathode[18,47]. For the densely packed secondary particles, the formation of intergranular cracks also contributes to the degradation of the cell. The general features of these intergranular cracks are representatively identified in the cross-sectional scanning electron microscopy (SEM) images in Fig. 1e–g, Fig. 2a,b and Supplementary Fig. 1. Comparing the pristine and cycled samples, it is obvious that after 100 cycles, the samples cycled at different high cutoff voltages of 4.2, 4.5, and 4.7 V exhibit no significant differences in terms of the intergranular cracking features.

Intragranular cracks are one of the significant differences for the samples cycled at different charge cutoff voltages. In the sample cycled at 4.7 V, the number of intragranular cracks were significantly higher than the samples cycled at 4.2 and 4.5 V, as shown representatively in Fig. 1e–g. Intragranular cracks are hardly seen in the samples cycled at 4.2 and 4.5 V, but are universally observed in the sample cycled at 4.7 V. In the sample cycled at 4.7 V, it would be expected that the intragranular cracking characteristics would substantially contribute to the

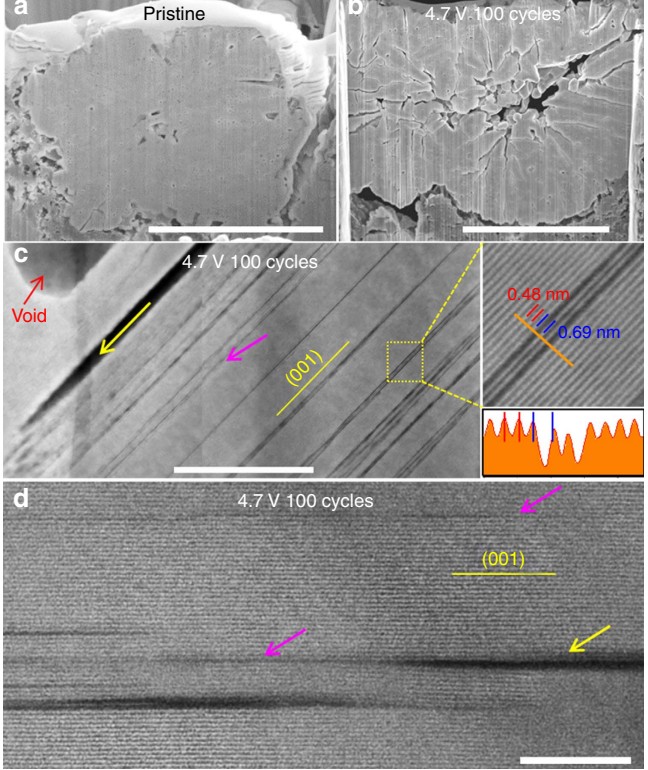

**Figure 2 | Intergranular and intragranular cracks.** Cross-sectional SEM images of secondary particles from (**a**) the pristine material and (**b**) the cycled one (100 cycles at the high cutoff voltage of 4.7 V). (**c**) and (**d**) are HAADF images from cycled $LiNi_{1/3}Mn_{1/3}Co_{1/3}O_2$ cathode particles, showing intragranular cracks along (001) plane. The yellow arrows indicate real cracks and the pink arrows indicate incubation cracks. Scale bars, 5 μm (**a,b**); 50 nm (**c**); and 10 nm (**d**).

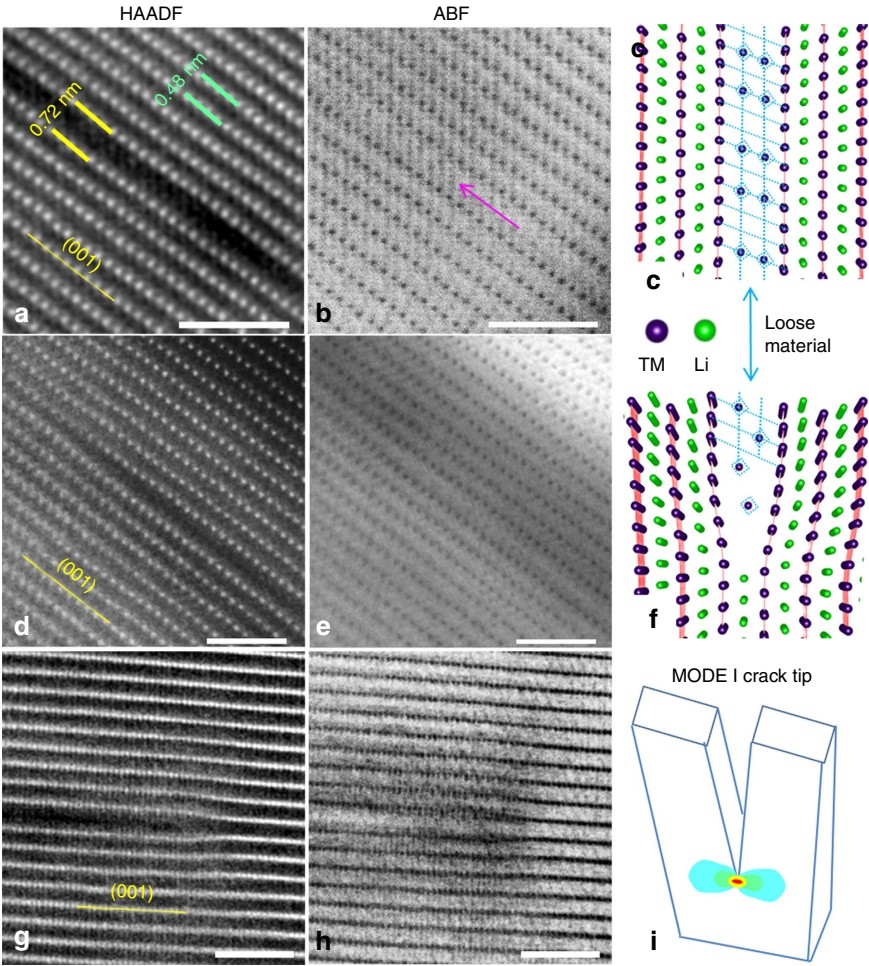

**Figure 3 | Lattice images of premature cracks.** Each pair of HAADF and ABF images are taken simultaneously. (**a,b**) [010] axis. (**c**) The corresponding lattice model. (**d,e**) A crack tip; (**f**) The corresponding model. (**g,h**) [1–10] axis. (**i**) Strain map at Mode I crack tip, which matches the strain contrast in **h**. Scale bars, 2 nm.

faster capacity degradation as compared with those cycled at 4.2 and 4.5 V. The abrupt increase of the density of intragranular cracks also indicates the high cutoff cycle voltage is the direct driving force for intragranular crack generation, suggesting a critical cycle voltage between 4.5 and 4.7 V for initializing the intragranular cracks in NMC333.

**Structural features of the intragranular cracks**. SEM and STEM observations reveal the details of these intergranular cracks as marked by the yellow arrows in Figs 1g and 2c, d. As the intra-granular cracks are mostly observed in the sample cycled at 4.7 V, we focused our effort on the NMC333 sample that was subjected to 100 cycles at 4.7 V in order to reveal the detailed structure of the intragranular cracks and understand their formation mechanism.

Two types of intragranular cracks can be uniquely identified. One type possesses the classical term of crack, which is featured by two free surfaces as indicated by the yellow arrows in the STEM high-angle annular dark-field (HAADF) images of Fig. 2c, d. The two free surfaces appear to be parallel along the whole crack except at the very tip, which is markedly different from a wedge-shaped crack formed by fast extension of crack under stress. Furthermore, cracks are predominantly parallel to (003) planes in the layer structure. These morphological features of the cracks are associated with their formation

process, which will be discussed in detail in the subsequent sections.

The other type of crack appears as narrow, dark strips when observed under STEM-HAADF imaging as indicated by the pink arrows in Fig. 2c, d. The dark strips are all parallel to the (003) planes (the layers) and distribute randomly with various spacing among the strips. The closest distance between two strips is a single layer of TM. As shown in the inset of Fig. 2c, the (003) plane spacing is 0.48 nm, while the dark contrasted strip corresponds to a widened TM plane spacing, ranging from 0.6 to 0.8 nm. Therefore, these dark contrasted strips appear to be formed by a parallel splitting of two adjacent TM layers, leading to a wider $c$ plane spacing.

It should be noted that, not always, but for some cases, the dark contrasted strip is spatially connected to the real crack as representatively shown in Fig. 2d. This observation likely indicates that the dark contrasted strip is a premature crack. The reason we term the dark contrasted strip as a premature crack is because it does not have two free surfaces. With the continued cycling of the battery, the premature crack will further develop into a real crack.

It is interesting to note that the dark strip contrasted region still possesses internal structural features. As shown in Fig. 3a,b and Supplementary Fig. 2, high-resolution STEM-HAADF and annular bright-field (ABF) images were simultaneously collected from one premature crack. As verified by the ABF image, the dark

contrasted strip that appeared in the STEM-HAADF image is actually not empty. As denoted by the pink arrow in Fig. 3b, there are some black dots that appeared with a rock-salt-like structure. Simulated HAADF/ABF images are shown in Supplementary Fig. 3 to support our interpretation. Thus, the dark contrasted strip, in fact, still contains an internal structure with material of low density. Its crystal model is illustrated in Fig. 3c. The very tip of the dark contrasted strip shows bending of the TM atomic row, forming a 'V' shape that looks just as the configuration of a crack tip. Fig. 3d, e are imaged from [010] axis and Fig. 3g, h are from [1–10] axis. The crystal model of the dark contrasted strip is illustrated in Fig. 3f. The ABF image shown in Fig. 3h even shows a strain contour at the tip, which is very similar to the strain contour of a real crack generated by tensile stress (Mode I crack, Fig. 3i). These structural features clearly demonstrate that the dark contrasted strips are premature cracks, which were formed by splitting the two neighbouring TM slabs and propagated along (003) planes.

Another significant feature of the intragranular cracking is that a large fraction of the intragranular cracks terminate within the grain interior, as representatively shown in Fig. 4, for which the red arrows highlight the intragranular end-to-end cracks that were fully terminated within the grain interior. These observations indicate that the intragranular cracks are initiated from the grain interior, which is in contrast with cracking models that predict the surface or grain boundary should be the preferred crack initiation site[42,44–46]. However, based on thermal analogy analysis, Kalnaus et al.[48] predicted cracking may initiate from the centre of the particle. Operando X-ray diffraction measurement has indicated the inhomogeneous lithiation/delithiation within a single cathode crystal[49]. Therefore, the variation of the $c$ plane spacing at different delithiation states within a single grain can lead to a complex strain pattern within the grain interior. Figure 4e gives a general illustration on the intragranular crack formation process under tensile stress. Inhomogenous Li distribution is believed to be the direct cause of such tensile stress. This internal cracking model also matches our proposed Mode I crack mechanism based on the observations on the crack tips in Fig. 3.

**Generation of dislocations in the primary particles.** Within the densely packed secondary particles, dislocation activity in the primary particle is another unique feature. High density of dislocations in both pristine and cycled samples were observed based on STEM-BF imaging, as representatively shown in Fig. 5a,b, for which the blue arrows highlight the dislocations. The dislocation density is in the range of $10^{11} \, \text{cm}^{-2}$. The observation of high-density dislocation in the primary particles is in marked contrast to the case of using nano-sized particles to assemble the battery electrode, where dislocation activity is hardly visible within the layer-structured particles. The high-density dislocations in the primary particle, as indicated in Fig. 5a,b, is the consequence of the formation of secondary particles by packing the smaller primary particles. Within the densely packed secondary particles, the primary particles are randomly oriented and in direct contact with their adjacent primary particles. Therefore, thermally (material synthesis process) and electrochemically (battery cycling) induced strain between neighbouring primary particles cannot be concordantly accommodated, which can inherently initiate dislocation in the primary particles. Figure 5c is a high-resolution STEM-HAADF image showing an end-on edge dislocation with Burger's vector $\vec{b} = 1/3[1\bar{1}01] \approx 0.5 \, \text{nm}$, which is a whole dislocation as shown in Fig. 5e. At the dislocation core area, due to the lattice mismatch and distortion, strain can build up accordingly. Our geometric phase analysis (GPA)[50] on Fig. 5c

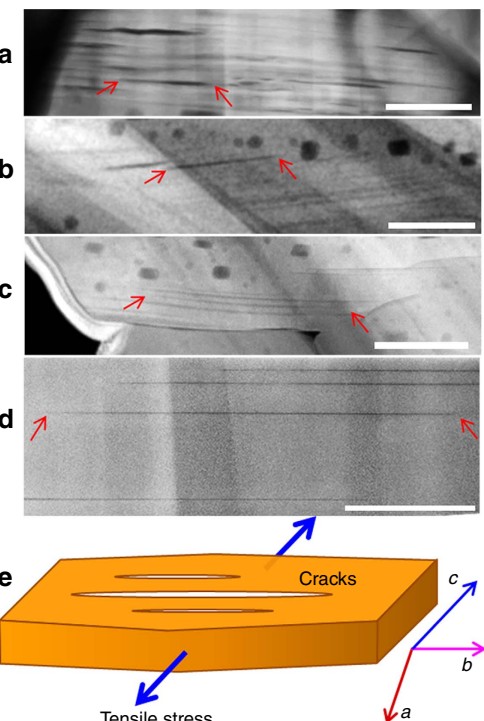

**Figure 4 | Intragranular cracks in LiNi$_{1/3}$Mn$_{1/3}$Co$_{1/3}$O$_2$ (NMC333).** The NMC333 particles are cycled 100 times with the high cutoff voltage of 4.7 V. In **a–d**, the arrows highlight the crack tips that are terminated in grain interior, and (**e**) a schematic diagram showing crack formation in the grain interior due to tensile stress. Scale bars, 100 nm (**a,b**); 200 nm (**c**); and 50 nm (**d**).

is shown in Fig. 5d. It can be seen in this out-of-plane strain map ($\varepsilon_{yy}$) that the left side with the extra plane is under compressive lattice strain, while the right side is under tensile strain.

The effects of dislocation on the battery properties can be evaluated from the following two aspects: firstly, the role of the dislocation itself on the ionic transport characteristics; and secondly, the evolution of dislocations and their effect on the structural stability of the material. In terms of dislocation itself, it is known that the dislocation core can act as a fast channel for ionic transport[49]. At the same time, the strain field associated with the dislocation can affect the active ion distribution and transport characteristics in the lattice. As shown in the GPA $\varepsilon_{yy}$ strain map in Fig. 5d, the lattice strain field introduced by the edge dislocation goes well beyond the dislocation core (indicated by the yellow arrow in Fig. 5c); in fact, previous Bragg coherent diffraction imaging[49] shows the strain field of an edge dislocation can reach more than 100 nm. Therefore, the high-density dislocations in the primary particles will definitely affect, either detrimentally or beneficially, the properties of cathode materials[41,49,51]. To our best knowledge, this is the first report on discovering the high-density dislocations in layered cathodes when the primary particles are packed as dense secondary particles.

**Discussion**
There have been many studies on the cracking mechanisms of cathode materials for LIBs[32,33,44–46,51,52]. However, these research efforts are mostly based on theoretical modelling of the stress-strain evolution. Fundamental understanding on the crack incubation is still far from clear. On the basis of what we

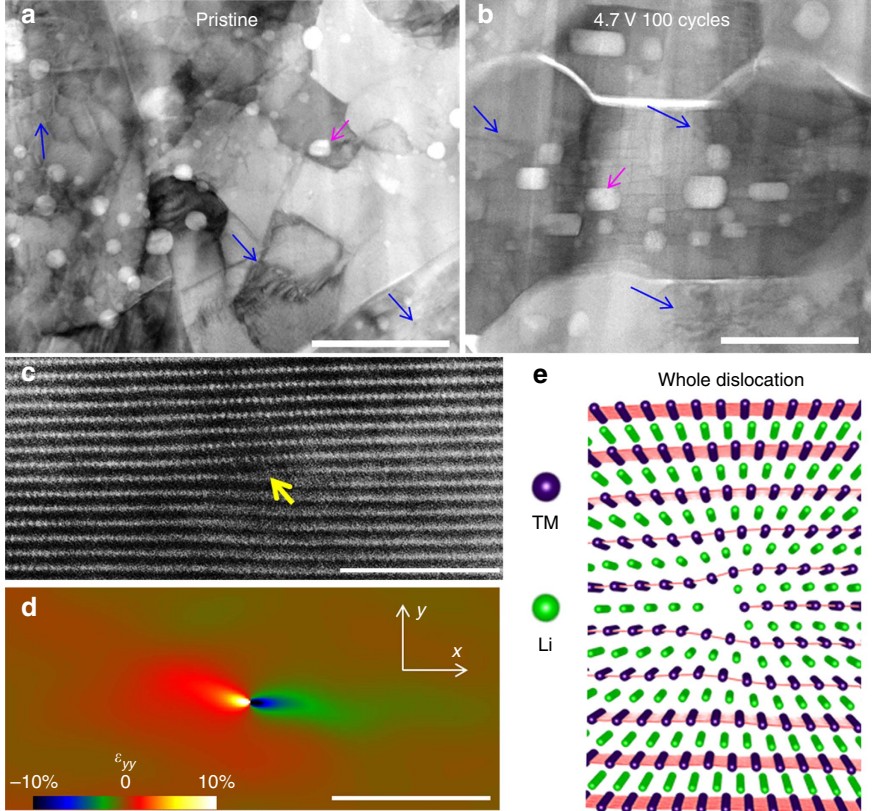

**Figure 5 | Dislocations in both pristine and cycled LiNi$_{1/3}$Mn$_{1/3}$Co$_{1/3}$O$_2$ (NMC333).** High density of dislocations are shown in the bright-field images of (**a**) pristine and (**b**) cycled NMC333 (after 100 cycles at the high cutoff voltage of 4.7 V). (**c**) A HAADF image showing an end-on edge dislocation in pristine NMC333, (**d**) is the corresponding strain map by GPA and (**e**) shows the dislocation model of (**c**). Scale bars, 200 nm (**a,b**); 5 nm (**c,d**).

have observed, the dark contrast strip (the premature crack) is the predecessor of the intragranular crack. The transition from the dark contrasted strip to the crack is a diffusion-controlled process, which is in essence an electrochemical driving process. Now, the key question is the origin of the dark contrasted strip or the premature cracks. On the basis of intensive observation using high-resolution STEM-HAADF imaging, we found that there exists a close correlation between edge dislocations and premature crack, as typically shown in Fig. 6. Fig. 6a and b show the nucleation of a premature crack at the dislocation core as indicated by the red arrows. Fig. 6d–f shows the association of an edge dislocation with premature cracks. These observations indicate that edge dislocation core can act as the nucleation site for crack incubation. From an energy point of view, Li and O ions will be preferentially removed from the tensile part of the dislocation core region to release strain. Kinetically, the dislocation core is a fast diffusion pathway. Therefore, the nucleation of the premature crack from the dislocation core is an electrochemically driven, but diffusion limited, process as schematically shown in Fig. 6c, which is similar to the fatigue-induced cracks in the slip band[53]. Our proposed mechanism is also supported by previous theoretical calculation work done by Huang *et al.*[51] who proposed dislocation-based cracking models.

We have drawn the conclusion that high-voltage cycling is the direct driving force for intragranular crack generation as evidenced by the drastic increase of intragranular crack density in the 4.7 V sample (Fig. 1g). Higher cycle voltage will result in deeper Li-ion extraction, which, on one hand, can aggravate structure instability, and on the other hand, can amplify the internal strain within a grain. Therefore, when the cycle voltage exceeds a critical value, in this case, some point between 4.5 and

4.7 V, intragranular cracks can be massively initiated as shown in Fig. 7a. As dislocation can act as a nucleation site for incubation of intragranular crack, we schematically illustrate in Fig. 7b the overall formation process of intragranular cracking based on a dislocation-nucleation mechanism. One of the fundamental questions is how the battery cycling rate contributes to the generation of intragranular cracks. To test the rate effect, the battery was cycled at 1C (as compared with 0.1C) at a cutoff voltage of 4.2 and 4.5 V. No premature cracks were identified at these samples. The results clearly point out the effect of cycle voltage on the intragranular crack formation.

The present observation has relevant implications for electrode design and battery operation. Although many surface coating methods have been used to minimize the surface-initiated structure degradation and cation dissolution in layer-structured cathode materials, these methods cannot be used to prevent the intrinsic, intragranular cracks that were initiated inside the primary particles and aggravated by the high-voltage charge process. During the Li-ion extraction and insertion processes, the lattice of crystal will be subject to change (either expansion or contraction). Although these changes are reversible within the certain limit, too large a lattice change, such as those induced by a high-voltage charge process, will lead to the irreversible formation of the dislocations and cracks, which will in turn be detrimental for the performance of the battery, as what has been observed in the present case. Therefore, on one hand, for the currently available NMC materials, the charge voltage has to be well controlled to minimize the electrochemically induced intragranular cracks. On the other hand, in order to push the NMC-layer-based materials for high-voltage applications, efforts have to be made to adjust the chemistry and structure of the material such

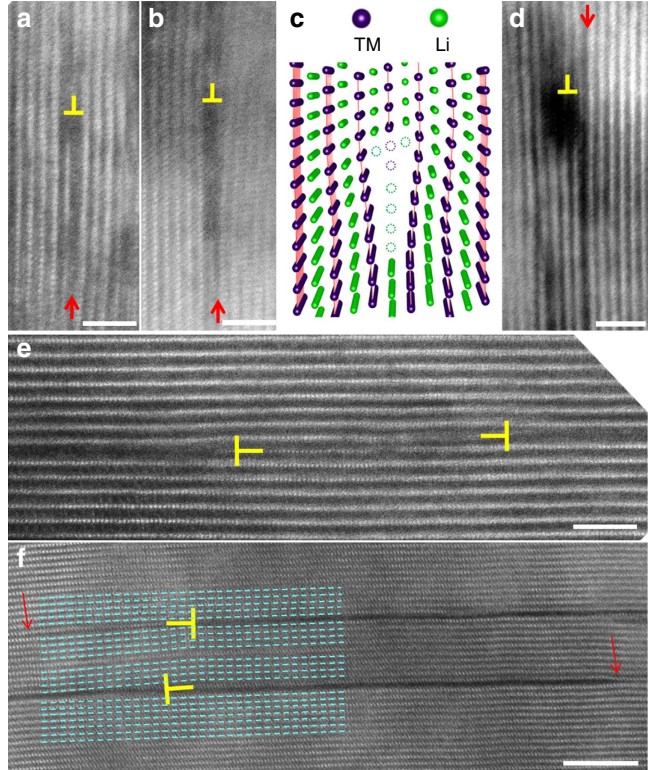

**Figure 6 | Dislocation associated with cracks in cycled LiNi$_{1/3}$Mn$_{1/3}$Co$_{1/3}$O$_2$.** (**a,b**) are the early incubation stages, showing vacancy condensation at dislocation core and (**c**) is the corresponding model. (**d–f**) show dislocations associated with cracks. Red arrows indicate crack tips. Scale bars, 2 nm; except **f** (5 nm).

that it can alleviate the internal grain strain, cause minimal Li distribution inhomogeneity, and retain a stable lattice during charge and discharge cycling.

## Methods

**Cathode material and cell test.** NMC333 pristine electrode laminates were provided by the Cell Analysis, Modelling, and Prototyping Facility at Argonne National Laboratory (pristine powders are commercially available and are manufactured by TODA KOGYO Company, Japan). The electrode laminates were punched into electrode disks that were ½ inches in diameter and dried at 75 °C overnight under a vacuum. Coin cells were assembled with the dried cathode electrode, metallic lithium foil as an anode electrode, Celgard2500 polyethylene (PE) membrane as separator, and 1 M lithium hexafluorophosphate (LiPF$_6$) dissolved in ethylene carbonate and dimethyl carbonate (1:2 in volume) as an electrolyte in an argon-filled MBraun glovebox. All the cathode electrodes were cycled at C/10 rate (1C = 180 mA g$^{-1}$) in the voltage range of 2.7–4.2 V, 2.7–4.5 V, 2.7–4.7 V and 2.7–4.8 V.

**Microstructure characterization and simulation.** FIB/SEM imaging and TEM specimen preparation by FIB lift out were conducted on a FEI Helios DualBeam Focused Ion Beam operating at 2–30 kV. Firstly, 1.2 µm thick Pt layer (200 nm e-beam deposition followed by 1 µm ion beam deposition) was deposited on the particles to be lifted out to avoid Ga ion beam damage. After lift out, the specimen was thinned to 200 nm using 30 kV Ga ion beam. A final polishing was performed using 2 kV Ga ion to remove the surface damage layer and further thinning to electron transparency. After a 2 kV Ga ion polish, the surface damage layer was believed to be <1 nm (ref. 54). The FIB-prepared NMC333 samples were investigated by using a JEOL JEM-ARM200CF microscope at 200 kV. This microscope is equipped with a probe spherical aberration corrector, enabling sub-angstrom imaging using STEM-HAADF/ABF detectors. For STEM-HAADF imaging, the inner and outer collection angles of an annular dark-field detector were set at 68 and 280 mrad, respectively. For STEM-ABF imaging, the inner and outer collection angles are 10 and 23 mrad, respectively. [010] and [1–10] zone axis STEM-HAADF/ABF images are simulated by using the QSTEM[55], which is a suite of software for quantitative image simulation of electron microscopy images, including model building and TEM/STEM/CBED image simulation. The collection angles for HAADF and ABF are 68–280 mrads and 10–20 mrads, respectively. A probe size of 0.8 Å is used with 27.5 mrad as convergence angle at 200 kV.

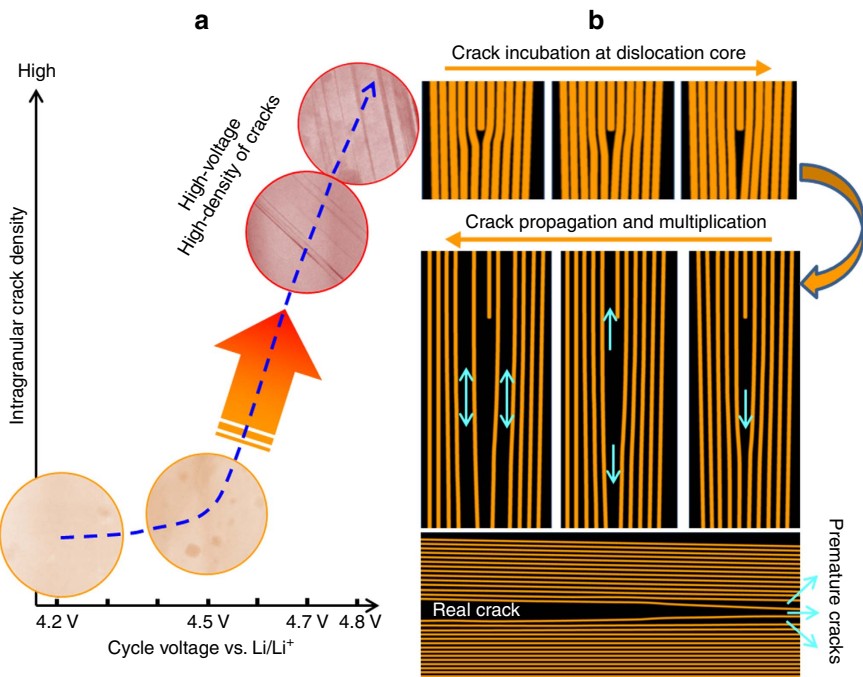

**Figure 7 | Cycle voltage governed intragranular cracking and underlying dislocation-based mechanism.** (**a**) HAADF images overlaid diagram shows the apparent dependence of intragranular cracking on the cycle voltage; when cycled below 4.5 V, intragranular crack can be hardly generated, while above 4.7 V, intragranular density shows a drastic increase; and (**b**) schematic diagrams to illustrate the dislocation-assisted crack incubation, propagation and multiplication process.

Different sample thicknesses (5, 10, 20, 30, 40 and 50 nm) with different focus values ($-5$, $-4$, $-3$, $-2$, $-1$, 0, 1, 2, 3, 4, 5 nm) are simulated.

**Data availability.** All relevant data are kept in storage at the Environmental Molecular Sciences Laboratory at Pacific Northwest National Laboratory and are available from the corresponding authors on request.

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

## Acknowledgements

We thank Dr Yuanyuan Zhu for help on the GPA analysis. This work was supported by the Assistant Secretary for Energy Efficiency and Renewable Energy, Office of Vehicle Technologies of the U.S. Department of Energy under Contract No. DE-AC02-05CH11231, Subcontract No. 6951379 under the Advanced Battery Materials Research (BMR) program. The microscopic analysis in this work was conducted in the William R. Wiley Environmental Molecular Sciences Laboratory (EMSL), a national scientific user facility sponsored by DOE's Office of Biological and Environmental Research and located at PNNL. PNNL is operated by Battelle for the Department of Energy under Contract DE-AC05-76RLO1830.

## Author contributions

C.-M.W., J.X., J.Z. and J-G.Z. initiated this research project. P.Y., J.Z., J.-G.Z. and C.-M.W. designed the experiment. J.Z. carried out material preparation and battery test. P.Y. conducted TEM experimental work and drafted the manuscript. All authors were involved in revising the manuscript.

## Additional information

**Competing financial interests:** The authors declare no competing financial interests.

