## [Peer Review File · Nature Communications]

Reviewers' comments:

Reviewer #1 (Remarks to the Author):

LiNi_{1/3}Mn_{1/3}Co_{1/3}O₂ (NMC333) is an important cathode material in LIB, and when it is cycled at high voltages, it experiences voltage fading which in turn causes capacity fading. Such capacity fading is usually attributed to the dissolution of transition metal cation into the electrolyte and decomposition of electrolyte. This paper presents another possible cause for the capacity fading problem: the intragranular crack. The HAADF data is convincing and well presented. Such results are obviously benefited from the recent advance of the aberration corrected electron microscopy technology. The results are of current interest. I recommend publication of the paper if the authors can address the following questions?

1. The authors suggest that the intragranular cracks are initiated from the grain interior, and such cracks are caused by dislocations in the grain interior, and the dislocations are caused by the inhomogeneous distribution of lithium. How is the lithium diffused into the grain interior? Is it an intercalation mechanism? And what causes the inhomogeneous lithium distribution in the grain interior?
2. Can the authors discuss whether the intragranular cracks are formed during charging or discharging process?
3. FIB is a heavy ion implantation process, and it modifies samples significantly. What kind of ion beam are being they used? Is it Ga ion beam? What is the voltage used? Need more details about the sample preparation procedure.
4. Are the samples cut along certain crystallographic orientation during the FIB process? If yes, how was the orientation of the sample determined?
5. What kind of software is used for the HAADF image simulation? Need more description about the image simulation.

Reviewer #2 (Remarks to the Author):

The authors presented a comprehensive post-mortem study of the intragranular cracking process in a layered LIB cathode material using high-resolution S/TEM. It has been shown that high-voltage cycling can lead to dislocation-assisted crack incubation and subsequent mode I crack

formation. This experimental work provides novel insights into understanding the mechanical degradation of layered cathode materials and offers an important guideline for rechargeable battery design. The manuscript is overall well-written and merits publication in Nature Communications. There are a few minor issues that the authors may consider addressing:

1. Figure 4e: The current schematic shows the region near the crack is Li rich and the region away from the crack is Li poor. Shouldn't it be the opposite? I would think a lower Li concentration near the crack region is required to induce a tensile stress and drive the crack growth.
2. The authors claim that the ABF image in Figure 3h recorded a strain map which was compared with a simulated strain map in Figure 3i. However, the image does not seem to have enough contrast to show the butterfly shaped deformation zone. The visual guidance of the yellow dashed line seems to be subjective.
3. Page 1, line 28: "maintain a structural stability" should be "maintaining ...".
4. Page 4, line 80: "Mitigation of intergranular cracking ..." should be "mitigation of intragranular ...".

Reviewer #3 (Remarks to the Author):

The manuscript describes the findings of transgranular cracks in NMC_{1,1,1} positive electrode material. The cracks are observed forming at very high upper cut-off potential for this material (4.7V). The report definitively shows the existence of such cracks supplemented by excellent microscopy. The novelty is in the fact that usually the cracking and degradation of NMC materials has been attributed to intergranular cracking resulting in separation of the primary particles. A model describing origination of cracks from edge dislocations nicely explains the findings.

A minor revision is recommended.

In Introduction, reference 42 is mentioned in context of surface coatings to reduce cracking (line 79, pg 4). The referenced work however deals with analysis and derivation of analytical solution for stresses in cylindrical electrode particle and it seems that it was mentioned erroneously.

While the accent in the manuscript is made on the upper cut-off potential as a major factor for crack formation, there is little or no mention of the influence of the Li removal rate on the crack formation. This should be important to consider. It seems that all cycling was done at C/10 rate. Would the similar transgranular cracks appear with lower cut-off voltage but much higher concentration gradients due to high charge/discharge rate?

"These observations indicate that the intragranular cracks must be initiated from grain interior. This phenomenon is against the applauded cracking models ...". Some models would place the crack on a particle surface (i.e. Woodford, Chiang, Carter, JES 157(10), 2010), while in others the maximum of damage and crack origin has been predicted for the particle core (Kalnaus, Rhodes, Daniel, J Power Sources 196(19) 2011) since this is where the maximum tensile stress will occur.

Please review the language carefully and correct the mistakes, for example "delitination" on page 7, or "high density dislocation in the primary particle" on page 9. The manuscript presents interesting findings and excellent microscopy analysis, but would benefit from improved language.

Manuscript NCOMMS-16-14632

“Atomistic process of intragranular cracking: a critical barrier for high-voltage usage of layer-structured cathode for lithium-ion batteries”

By Pengfei Yan, Jianming Zheng, Meng Gu, Jie Xiao, Ji-Guang Zhang, and Chong-Min Wang

Point by point reply to the reviewers' comments:

Reviewer #1 (Remarks to the Author):

LiNi_{1/3}Mn_{1/3}Co_{1/3}O₂ (NMC333) is an important cathode material in LIB, and when it is cycled at high voltages, it experiences voltage fading which in turn causes capacity fading. Such capacity fading is usually attributed to the dissolution of transition metal cation into the electrolyte and decomposition of electrolyte. This paper presents another possible cause for the capacity fading problem: the intragranular crack. The HAADF data is convincing and well presented. Such results are obviously benefited from the recent advance of the aberration corrected electron microscopy technology. The results are of current interest. I recommend publication of the paper if the authors can address the following questions?

Response: Thanks to the review for the great positive comment of our work

Question 1. The authors suggest that the intragranular cracks are initiated from the grain interior, and such cracks are caused by dislocations in the grain interior, and the dislocations are caused by the inhomogeneous distribution of lithium. How is the lithium diffused into the grain interior? Is it an intercalation mechanism? And what causes the inhomogeneous lithium distribution in the grain interior?

Response to Question 1: Yes, lithium diffusion is an intercalation mechanism. For NMC333 cathode, lithium is diffused along Li-layers (2D diffusion). The diffusion pathway is believed via oxygen octahedron-tetrahedron-octahedron. (1)

The inhomogeneous lithium distribution in cathode materials has been experimentally observed previously. (2-4) Reasons that cause lithium inhomogeneous distributions can be electrical conductivity, strain/stress, defects, and composition fluctuation. In our case, we believe strain/stress and defects are the main factors that cause inhomogeneous lithium distribution in the grain interior.

- (1) Kang K, Meng YS, Breger J, Grey CP, Ceder G. Electrodes with high power and high capacity for rechargeable lithium batteries. *Science* 311, 977-980 (2006).
- (2) Holtz, M. E.; Yu, Y.; Gunceler, D.; Gao, J.; Sundararaman, R.; Schwarz, K. A.; Arias, T. A.; Abruña, H. D.; Muller, D. A. Nanoscale Imaging of Lithium Ion Distribution During In Situ Operation of Battery Electrode and Electrolyte. *Nano Lett.* 2014, 14, 1453-1459.
- (3) Ulvestad, A.; Singer, A.; Clark, J. N.; Cho, H. M.; Kim, J. W.; Harder, R.; Maser, J.; Meng, Y. S.; Shpyrko, O. G. *Science* 2015, 348, (6241), 1344-1347.

(4) Gent, W. E.; Li, Y.; Ahn, S.; Lim, J.; Liu, Y.; Wise, A. M.; Gopal, C. B.; Mueller, D. N.; Davis, R.; Weker, J. N.; Park, J.-H.; Doo, S.-K.; Chueh, W. C. *Adv. Mater.* 2016, 28, (31), 6631-6638.

Question 2. Can the authors discuss whether the intragranular cracks are formed during charging or discharging process?

Response to Question 2: This is a very good question. Actually, as discussed in the text, we believe the formation of intragranular cracks, especially the premature ones, is a gradual process, which is contributed by the process during both charging and discharging. Fundamentally, cracks are generated to release strain, therefore, as long as strain is introduced and is large enough, there is a possibility to form a crack. In both charging and discharging processes, strain can be introduced. Therefore, charging and discharging are more likely acting as a fatigue process to facilitate crack formation.

Question 3. FIB is a heavy ion implantation process, and it modifies samples significantly. What kind of ion beam are being they used? Is it Ga ion beam? What is the voltage used? Need more details about the sample preparation procedure.

Response to Question 3: For FIB cutting of the sample, Ga ion beam was used. We agree that Ga ion can damage the sample during sample preparation, but using lower voltage can significantly reduce the thickness of surface damage layer. In our case, we used 2kV (current: 25pA) to make the final polishing. The surface damage layer is believed to be less than 1nm. (1) As a contrast, a sample is usually >50nm in thickness, therefore, the surface damage layer is negligible. **The detailed procedure and parameters for FIB lift out have been added in the revised version.**

(1) Mayer, J.; Giannuzzi, L. A.; Kamino, T.; Michael, J. *MRS Bull.* 2011, 32, (5), 400-407.

Question 4. Are the samples cut along certain crystallographic orientation during the FIB process? If yes, how was the orientation of the sample determined?

Response to question 4: The secondary particle is an agglomeration of the primary particles, so it is not necessary to cut along a certain crystallographic direction. We just randomly cut the secondary particle and then do lift-out. Usually, we can get several primary particles thin enough (<100 nm) and then we tilt the sample to certain zone axis to acquire STEM-HAADF/ABF lattice images. **We provided a series of images in Supplementary Fig. 4 to show a TEM specimen preparation process.**

Question 5. What kind of software is used for the HAADF image simulation? Need more description about the image simulation.

Response to Question 5: We used QSTEM to do HAADF/ABF image simulations. **Details of the simulation parameters has been added in the revised version.**

Reviewer #2 (Remarks to the Author):

The authors presented a comprehensive post-mortem study of the intragranular cracking process in a layered LIB cathode material using high-resolution S/TEM. It has been shown that high-voltage cycling can lead to dislocation-assisted crack incubation and subsequent mode I crack formation. This experimental work provides novel insights into understanding the mechanical degradation of layered cathode materials and offers an important guideline for rechargeable battery design. The manuscript is overall well-written and merits publication in Nature Communications. There are a few minor issues that the authors may consider addressing:

Response: Thanks to the reviewer for the great comment to our work.

Question 1. Figure 4e: The current schematic shows the region near the crack is Li rich and the region away from the crack is Li poor. Shouldn't it be the opposite? I would think a lower Li concentration near the crack region is required to induce a tensile stress and drive the crack growth.

Response to Question 1: We agree with reviewer's comment that Li ion must be moved away to facilitate crack's propagation. Because the direct driving force of cracking is tensile strain. For NMC333, c-plane spacing will expand to maximum when 60% Li extracted out and then it will shrink with further delithiation,⁽¹⁾ which means tensile stress can be built up at both Li-poor and Li-rich situations. We modified Fig. 4e in the revision.

(1) Dolotko, O.; Senyshyn, A.; Mühlbauer, M. J.; Nikolowski, K.; Ehrenberg, H. Understanding structural changes in NMC Li-ion cells by in situ neutron diffraction. *J. Power Sources* 2014, 255, 197-203.

Question 2. The authors claim that the ABF image in Figure 3h recorded a strain map which was compared with a simulated strain map in Figure 3i. However, the image does not seem to have enough contrast to show the butterfly shaped deformation zone. The visual guidance of the yellow dashed line seems to be subjective.

Response to question2: Thanks for the comment. We agree the strain contrast from ABF image is not strong enough, but still can be recognized. Therefore, we prefer to keep this image in the revision. In order to avoid a subjective guidance, we have removed the dashed line from Fig. 3h and the image contrast was adjusted to highlight the strain region at the crack tip.

Question 3. Page 1, line 28: "maintain a structural stability" should be "maintaining ...".

Question 4. Page 4, line 80: "Mitigation of intergranular cracking ..." should be "mitigation of intragranular ...".

Response to Questions 3 &4: Thanks for the corrections. They have been corrected in the revision.

Reviewer #3 (Remarks to the Author):

The manuscript describes the findings of transgranular cracks in NMC_{1,1,1} positive electrode material. The cracks are observed forming at very high upper cut-off potential for this material (4.7V). The report definitively shows the existence of such cracks supplemented by excellent microscopy. The novelty is in the fact that usually the cracking and degradation of NMC materials has been attributed to intergranular cracking resulting in separation of the primary particles. A model describing origination of cracks from edge dislocations nicely explains the findings.

A minor revision is recommended.

Response: Thanks to the reviewer for the great comment of our work.

Question 1. In Introduction, reference 42 is mentioned in context of surface coatings to reduce cracking (line 79, pg 4). The referenced work however deals with analysis and derivation of analytical solution for stresses in cylindrical electrode particle and it seems that it was mentioned erroneously.

Response to Question 1: Thanks for the correction. We made a mistake on that and it has been replaced by the right reference “Tan, G.; Wu, F.; Li, L.; Chen, R.; Chen, S. J. Phys. Chem. C 2013, 117, (12), 6013-6021.”

Question 2: While the accent in the manuscript is made on the upper cut-off potential as a major factor for crack formation, there is little or no mention of the influence of the Li removal rate on the crack formation. This should be important to consider. It seems that all cycling was done at C/10 rate. Would the similar transgranular cracks appear with lower cut-off voltage but much higher concentration gradients due to high charge/discharge rate?

Response to Question 2: This is a very good question. We actually have started the investigations on the effect of cycle rate. Per the reviewer’s suggestion, we observed 100 cycles NMC333 that cycled 4.2 V and 4.5V at 1C rate (as compared with 0.1C). We do not see any premature intragranular cracks in the 1C cycled sample. The result confirms our conclusion that the voltage plays major role on the intergranular cracking. This point has been reflected in the revised manuscript.

Question 3: "These observations indicate that the intragranular cracks must be initiated from grain interior. This phenomenon is against the applauded cracking models ...". Some models would place the crack on a particle surface (i.e. Woodford, Chiang, Carter, JES 157(10), 2010), while in others the maximum of damage and crack origin has been predicted for the particle core (Kalnaus, Rhodes, Daniel, J Power Sources 196(19) 2011) since this is where the maximum tensile stress will occur.

Response to Question 3: Thanks for the comments and information. The statement has been modified and Kalnaus’s work has been cited in the revision.

Question 4: Please review the language carefully and correct the mistakes, for example

“delitination” on page 7, or “high density dislocation in the primary particle” on page 9. The manuscript presents interesting findings and excellent microscopy analysis, but would benefit from improved language.

Response to question 4: Thanks for the comments and suggestion. The PNNL media team has helped to fix the language upon revision.

Reviewers' Comments:

Reviewer #1 (Remarks to the Author):

My questions have been addressed in the revised MS. I recommend publication of the MS in its present form.

Reviewer #3 (Remarks to the Author):

It's a very interesting report and technically sound. The authors addressed all the comments from the previous review. The fact that the transgranular cracking depends only on the upper cut-off potential and not on the applied current density is intriguing.

The manuscript can be accepted with some minor corrections that don't affect any technical content and can be implemented at the proofreading stage:

Pg. 4, lines 77-78: "... they can generate much more fresh surfaces ..." either delete "much" or replace it with "many".

Pg. 4, line 83: "...and carefully controls of cycle conditions." Replace with "careful controls of cycle conditions."

Pg. 8, line 197: "The high density dislocations in the primary particle ...", - probably the authors meant "high dislocation density" or "high density of dislocations". The same goes for line 218 on Pg. 9.

Pg. 7, line 183: "delitination"? Probably the authors meant "delithiation".

Pg. 7, line 180. As far as I can tell the authors of the reference 48 proposed a damage model to predict cracking from the particle center. The "thermal analogy" approach was used, but not first proposed by them. Please either delete this reference or modify the sentence.

Reviewer #1 (Remarks to the Author):

My questions have been addressed in the revised MS. I recommend publication of the MS in its present form.

Reviewer #3 (Remarks to the Author):

It's a very interesting report and technically sound. The authors addressed all the comments from the previous review. The fact that the transgranular cracking depends only on the upper cut-off potential and not on the applied current density is intriguing.

The manuscript can be accepted with some minor corrections that don't affect any technical content and can be implemented at the proofreading stage:

Pg. 4, lines 77-78: "... they can generate much more fresh surfaces ..." either delete "much" or replace it with "many".

Pg. 4, line 83: "...and carefully controls of cycle conditions." Replace with "careful controls of cycle conditions."

Pg. 8, line 197: "The high density dislocations in the primary particle ...", - probably the authors meant "high dislocation density" or "high density of dislocations". The same goes for line 218 on Pg. 9.

Pg. 7, line 183: "delitination"? Probably the authors meant "delithiation".

Response: Above typos and grammar errors have been corrected.

Pg. 7, line 180. As far as I can tell the authors of the reference 48 proposed a damage model to predict cracking from the particle center. The "thermal analogy" approach was used, but not first proposed by them. Please either delete this reference or modify the sentence.

Response: The sentence has been modified according to the reviewer's suggestion